# Deworming and micronutrient status by community open defecation prevalence: An observational study using nationally representative data from India, 2016–2018

Suman Chakrabarti[1], Sitara S. R. Ajjampur[2], Hugh Sharma Waddington[3], Avinash Kishore[1], Phuong H. Nguyen[1], Samuel Scott[1] *

1 International Food Policy Research Institute, Washington DC, and India, 2 The Wellcome Trust Research Laboratory, Christian Medical College, Vellore, India, 3 London School of Hygiene & Tropical Medicine and London International Development Centre, London, United Kingdom

* Samuel.Scott@cgiar.org

## Abstract

### Background

Micronutrient deficiencies are widespread in India. Soil-transmitted helminth (STH) infections are acquired by interaction with soil and water contaminated by human feces and lead to blood loss and poor micronutrient absorption. The current recommendation for control of STH-related morbidity is targeted deworming, yet little is known about the effectiveness of deworming on micronutrient status in varying sanitation contexts. Ranging between 1% and 40% prevalence across Indian states, open defecation (OD) remains high despite India's investments at elimination by promoting community-wide sanitation. This variation provides an opportunity to study the relationship between deworming, micronutrient status, and OD at-scale.

### Methods and findings

Cross-sectional datasets that were representative for India were obtained the Comprehensive National Nutrition Survey in 2016 to 2018 ($n = 105,060$ individuals aged 1 to 19 years). Consumption of deworming medication was described by age and community OD level. Logistic regression models were used to examine the relationship between deworming, cluster OD, and their interactions, with anemia and micronutrient deficiencies (iron, zinc, vitamin A, folate, and vitamin B12), controlling for age, sex, wealth, diet, and seasonality. These regression models further allowed us to identify a minimum OD rate after which deworming becomes ineffective. In sensitivity analyses, the association between deworming and deficiencies were tested in subsamples of communities classified into 3 OD levels based on statistical tertiles: OD free (0% of households in the community practicing OD), moderate OD (>0% and <30%), or high OD (at least 30%). Average deworming coverage and OD prevalence in the sample were 43.4% [IQR 26.0, 59.0] and 19.1% [IQR 0, 28.5], respectively. Controlling for other determinants of nutritional status, adolescents living in

**Data Availability Statement:** All relevant data are within the paper and its Supporting Information files.

**Funding:** SSRA is supported by the NIH FIC Emerging Global Leader Award (K43TW011415). The funders had no role in study design, data collection and analysis, decision to publish, or preparation of the manuscript.

**Competing interests:** The authors have declared that no competing interests exist.

**Abbreviations:** CNNS, Comprehensive National Nutrition Survey; LMIC, low- and middle-income country; MoHFW, Ministry of Health and Family Welfare; OD, open defecation; OR, odds ratio; PCA, principal component analysis; PSU, primary sampling unit; SBM, Swachh Bharat Mission; STH, soil-transmitted helminth.

communities with higher OD levels had lower coverage of deworming and higher prevalence of anemia, zinc, vitamin A, and B12 deficiencies. Compared to those who were not dewormed, dewormed children and adolescents had lower odds of anemia (adjusted odds ratio 0.72, (95% CI [0.67, 0.78], $p < 0.001$) and deficiencies of iron 0.78, (95% CI [0.74, 0.82], $p < 0.001$) and folate 0.69, (95% CI [0.64,0.74], $p<0.001$)) in OD free communities. These protective effects remained significant for anemia but diminished for other micronutrient deficiencies in communities with moderate or high OD. Analysis of community OD indicated a threshold range of 30% to 60%, above which targeted deworming was no longer significantly associated with lower anemia, iron, and folate deficiency. The primary limitations of the study included potential for omitted variables bias and inability to capture longitudinal effects.

## Conclusions

Moderate to high rates of OD significantly modify the association between deworming and micronutrient status in India. Public health policy could involve sequencing interventions, with focus on improving deworming coverage in communities that have achieved minimum thresholds of OD and re- triggering sanitation interventions in high OD communities prior to deworming days, ensuring high coverage for both. The efficacy of micronutrient supplementation as a complementary strategy to improve nutritional outcomes alongside deworming and OD elimination in this age group needs further study.

---

### Author summary

#### Why was this study done?

- Deworming is recommended to tackle soil-transmitted worm infections in low- and middle-income countries (LMICs), but the effectiveness of deworming provided on a large scale varies significantly.

- Sanitation levels across communities are likely to influence the impact of deworming in populations due to high rates of reinfection, but no studies have investigated these links.

- Despite well-intended programs aiming to eliminate open defecation in India, this behavior persists in certain communities.

- This study was done to test whether higher open defecation levels diminished the protection provided by deworming against anemia or micronutrient deficiencies among Indian children and adolescents.

#### What did the researchers do and find?

- We used a cross-sectional dataset (sample size 105,060 individuals) representative of Indian children and adolescents to look at links between open defecation, deworming, anemia, and micronutrient deficiencies.

- We found that deworming coverage remains low in India, especially among young children and adolescents out of school, indicating the need for enhanced and expanded deworming interventions.

- After accounting for the influence of age, sex, wealth, education, diet, and season, dewormed children in areas without open defecation had lower rates of anemia (28% lower), iron deficiency (22% lower), and folate deficiency (31% lower); however, in communities with open defecation, dewormed children showed similar deficiency levels as those not dewormed.

### What do these findings mean?

- Our findings suggest that open defecation diminishes the protection provided by deworming medication against anemia and micronutrient deficiencies in India.

- The findings highlight the importance of sequencing sanitation interventions such as behavior change campaigns before deworming initiatives in high open defecation communities.

- This study cannot establish causation because the statistical models were based on observing associations at a single point in time rather than tracking changes over time.

## Introduction

Soil-transmitted helminth (STH) infections affect 875 million children worldwide annually, with most cases found in low- and middle-income countries (LMICs) [1]. These infections have been linked to detrimental health consequences, such as chronic blood loss leading to iron deficiency anemia and impaired nutrient absorption in the gut, resulting in malnutrition [1,2]. STH infections contribute to both micronutrient deficiencies and anemia among children, albeit with variations in pathophysiology based on the specific worm species involved [3,4]. Global deworming strategies recommend the routine administration of deworming medications to children aged 1 to 9 years and nonpregnant adolescents aged 10 to 19 years in regions with a high prevalence of STH infections [5]. However, it is crucial to recognize that the effectiveness of these targeted deworming programs varies significantly. This variability is influenced by the endemicity of infections, which in turn is dependent on the accessibility and utilization of water and sanitation facilities [6–8]. Evidence suggests that improved sanitation facilities can reduce the risk of all STH infections by 40% [9]. In contrast, unimproved sanitation—classified as situations where households can only access facilities that do not adequately isolate and treat feces, facilities shared with other households, or no facilities so that they defecate in the open—presents challenges in reducing malnutrition due to high worm reinfection rates [9]. Therefore, it may be important to consider the relative roles and timing of deworming alongside efforts to improve sanitation access at the household level (the necessary condition to reduce open defecation) and behavior change communication activities promoting sanitary practices (the sufficient condition), when designing public health programs that aim to reduce morbidity and worm burden in particular contexts.

India bears a substantial burden of STH infections, with over 200 million individuals affected by these parasitic afflictions [10]. The prevalence of roundworm (*Ascaris lumbricoides*), whipworm (*Trichuris trichiura*), and hookworm (*Ancylostoma* spp. and *Necator americanus*) infections among school-aged Indian children was 25%, 13%, and 10%, respectively, in 2022, albeit with large subnational heterogeneity [11]. Moreover, the burden of anemia in Indians <19 years stands at 67% [12]. The prevalence of micronutrient deficiencies including iron (54%), vitamin B12 (53%), and folic acid (37%) are also substantial [13]. Hookworm infections are a significant global contributor to anemia, predominantly impacting populations with low iron reserves—children and women of reproductive age. Roundworms can also induce anemia through mucosal bleeding in the upper gastrointestinal tract or through general inflammation [14]. Whipworm infections can also induce a local inflammatory response and blood loss (from bleeding and oozing at the mucosal entry sites of worms) [4,15]. While Ascaris has been reported to be the most prevalent STH infection globally, hookworm infections are linked to the highest DALYs and leads to iron deficiency anemia, hypoalbuminemia, and hypoproteinemia due to chronic blood loss especially in susceptible populations with low iron reserves such as pregnant women (leading to intra-uterine growth retardation and low birth weights) and young children (leading to stunting and cognitive deficits) and in heavy intensity infections [16]. Symptoms are also dependent on species with *A. duodenale* leading to 10 times as much blood loss as that caused by *N. americanus* [17]. Ascaris leads to malnutrition, vitamin A and C deficiency, cognitive deficits, and lack of physical fitness in children with high worm burdens due to anorexia and absorption defects [18]. Due to large worm size, this can also lead to acute intestinal and biliary tract obstruction. In trichuriasis, inflammation of large intestine and blood loss can lead to colitis (especially in heavy intensity infections), iron deficiency anemia, and stunting as well as diarrhea in children [18].

Numerous factors collectively contribute to the persistence of STH transmission cycles in India [19]. These factors include a high population density, inadequate personal hygiene practices, employment in agriculture and most importantly, the prevalence of open defecation (OD). In areas characterized by high rates of OD, the ingestion of roundworm and whipworm eggs through fecally contaminated food and water is a common route of transmission, while hookworms are transmitted through skin contact with fecally contaminated soil containing infectious larvae [20]. Despite India's investments in the Total Sanitation Campaign (started in 1999)—later renamed the Clean India or Swachh Bharat Mission (SBM) in 2014, with the broader aim of eliminating OD—by promoting the construction and utilization of toilets, and raising public awareness about cleanliness and hygiene, high levels of OD persist [21–23]. The SBM has led to the construction of over 500 million toilets in households, schools, and public areas [21]. As of 2021, more than 250 million individuals in India continue to practice OD, indicating the high levels of human fecal material in the environment, particularly in rural areas [22]. The OD burden is particularly high in the Gangetic planes and interior states (Uttar Pradesh, Bihar, Jharkhand, Odisha, Rajasthan), partly by choice, as OD is expressed as a medium for socializing, a habit, and an outdoor activity that complies with social and religious norms [24]. This ongoing presence of fecal material likely contributes to the continued transmission of STH infections.

To address the high STH burden among children, India initiated the National Deworming Day program in 2015, one of the largest national public health programs in the world [25–27], with the aim of deworming every child between 1 and 19 years of age biannually [20–22]. The Anemia Mukt Bharat (Anemia Free India) program, launched in 2018, added deworming of pregnant women in the second trimester [28]. However, evidence on the coverage and effectiveness of these targeted deworming programs in India is mixed [25]. Systematic reviews of global studies show inconclusive evidence on protective effects of deworming on anemia, and

no studies have investigated links between deworming and micronutrient status in India, despite a high burden of STH and micronutrient deficiencies [29,30]. It is also not clear whether the rigorous evidence that supports the effectiveness of deworming, primarily from randomized field trials in sub-Saharan Africa, is relevant for India [31–33]. These studies are usually conducted in subnational areas, with specific levels of exposure to OD and STH infections but have been very influential for global programmatic strategies [30,34]. Evidence from larger surveys, preferably at the national level, that can evaluate the associations between deworming and nutrition outcomes for different exposures to OD, would be useful to inform decision-making around population-level interventions to reduce disease exposure.

In this study, we hypothesize that the negative association between deworming and anemia or micronutrient deficiencies will move closer to the null with increasing community OD levels, due to significant STH exposure and re- infections in such areas. Given high burdens and variability in micronutrient deficiencies, OD, and deworming coverage, India provides an ideal context to test this hypothesis. In this paper, we: (1) examined the coverage of deworming among children aged 1 to 19 years; (2) examined associations between deworming and anemia or micronutrient deficiencies; and (3) assessed how the level of OD in the community modifies these associations.

## Methods

### Data

We used cross-sectional secondary data from the Comprehensive National Nutrition Survey (CNNS) published by the Ministry of Health and Family Welfare (MoHFW), Government of India, UNICEF and the Population Council [35]. Conducted between February 2016 and October 2018, the CNNS was the first nationally representative survey designed to examine a comprehensive set of outcomes and risk factors on the nutritional status of Indian children and adolescents. The CNNS followed a multi-stage sampling design and covered all Indian states. In the first stage, rural villages and urban neighborhoods (the primary sampling units, PSUs) were selected with probability proportional to size from a national list of primary census tracts. In the second stage, households were randomly selected within each PSU with a catchment area of 100 to 150 households. Households with members between 0 and 19 years were classified into 3 categories (0 to 4 years, 5 to 9 years, and 10 to 19 years) for selection from each group. In each household, only 1 participant was selected from each age group. For households with eligible participants from more than 1 age group, a maximum of 3 (1 from each group) participants were interviewed. Participants with a major physical disability, infectious illness, chronic systemic illness, injury, or pregnancy were excluded from the interview. Accounting for design effects and maximizing the sample power with the available resources, the planned sample size was 40,700 individuals in each of the 3 age groups for the household survey and anthropometric measurements and 20,350 individuals for biological samples for each of the 3 age groups. Our analytical sample comprised 105,060 children aged 1 to 19 years from 30 out of 37 Indian states and union territories and 2,035 communities (PSUs). Further details on sampling methodology can found in the CNNS report [35]. Parental consent and child assent were obtained for all children per Indian guidelines. Ethical approval was obtained by the Population Council's Institutional Review Board and by the Postgraduate Institute of Medical Education and Research, Chandigarh.

### Outcomes

The outcomes of interest were anemia and micronutrient deficiencies (including iron, zinc, vitamin A, folate, vitamin B12, and vitamin D). Individuals were classified deficient in a

micronutrient when their serum level observation fell below the threshold for deficiency (**S1 Table**) [35,36]. Blood was collected in the morning via antecubital venipuncture by 360 trained phlebotomists (qualified with a Diploma in Medical Laboratory Technology). Eight milliliters (ml) of blood were collected from children 1 to 4 years and 10 ml from children 5 to 19 years. Once collected, the vacutainer tubes with blood samples were placed in cool boxes without direct contact with the icepacks and transported to the nearest pre-identified collection center at appropriate temperatures. The blood samples for serum retinol were covered with aluminum foil soon after collection to protect against light exposure. At the collection center, the blood samples were spun for 20 min and aliquoted into appropriately sized tubes for laboratory testing and aliquots were stored for laboratory analysis (plasma at 5 to 7˚C, serum frozen) [29]. Biological analyses were conducted using appropriate techniques (**S1 Table**) at laboratories participated in the US CDC VITAL-EQA program [29,37].

## Exposures

The primary intervention of interest was consumption of deworming medication (hereafter referred to as "deworming") in the previous 6 months, reported by the parent for young children (aged 0 to 9) and self-reported by older children (aged 10 to 19 years). Source of obtaining deworming medication was also used for descriptive analysis. A secondary exposure of interest was OD, measured as the average proportion of sampled households practicing open defecation in a PSU (henceforth "community") [23]. In sensitivity analyses, we also constructed 3 OD contexts for ease of interpretation: OD free (OD = 0%), moderate OD (>0%–30% of households in a community practicing OD), and high OD (≥30%) for regression analysis. Thresholds for moderate and high OD divided the sample into the 50th and 75th percentiles.

## Covariates

Demographic covariates included were child age (year) and sex (female dummy). To control for socioeconomic position, paternal and maternal education (years), and household wealth were included in all models to account for large unobserved biasing factors such as income and access to health care, among others. Principal component analysis (PCA) was used to construct the asset-and-amenities-based household wealth index which was further categorized into quintiles [38]. Variables included in the PCA were primary fuel used for lighting, access to electricity, materials used in dwelling (floor, walls, and roof), house and land ownership, and possession of 26 durable assets [38,39].

CNNS used a pre-validated short food frequency questionnaire to collect data on the number of days per week individuals consumed any amounts of 17 food groups [40,41]. To account for diet, a significant predictor of micronutrient status [42], we constructed indicators which represented the number of food groups (at least weekly consumption of foods in a category) in 3 categories—animal-sourced foods (milk or milk products, eggs, fish, chicken, and meat), plant-sourced foods (pulses or beans, green leafy vegetables, other vegetables, fruits, and nuts/seeds), and unhealthy foods (fried foods (poori, pakora, vada, samosa, tikki), junk food (burger, pizza, pasta, instant noodles), sweets (Indian sweets, pastries/cakes, donuts), and sugar-sweetened beverages [43]). Within each of the 3 categories, we counted the number of food groups that were consumed in the previous week to proxy the intake of healthy and unhealthy foods. Given India's large vegetarian population, we split the healthy foods into 2 indicators, plant, and animal-based foods [44]. A lack of data on some food groups including dark leafy greens and vegetables, or Vitamin A-rich fruits and vegetables, precluded calculation of the standard dietary diversity indicator [45].

Season dummies for summer (March to May), monsoon (June to September), and post-monsoon (October to November) to account for differences in temperature and rainfall conditions across states in India that may drive infections and food availability. December to February was treated as the reference category.

## Statistical analysis

We performed 3 sets of statistical analyses. Firstly, we reported the prevalence (percent and 95% confidence intervals) of anemia and micronutrient deficiencies by child deworming status and community OD level. Secondly, we graphed the receipt of deworming medication by age and community OD level using local polynomial smoothing. We used Stata defaults for plotting the curves with the epanechnikov kernel, a rule-of-thumb bandwidth, and a zero-degree polynomial (local-mean smoothing). Thirdly, we used logistic regression to (1) examine the association between deworming and anemia or micronutrient deficiency outcomes; and (2) identify a minimum OD rate after which deworming becomes ineffective. Explanatory variables included in the model were deworming (binary), cluster OD (continuous ranging from 0, zero OD prevalence to 1, maximum OD prevalence), and their interaction. All other covariates were also included in these models to reduce confounding. All regressions were clustered at the PSU level to account for correlation of outcomes within communities. The interaction term models that the association of deworming with the outcome depends on the level of OD. A positive and significant interaction term implies that the benefits of deworming for the given outcome are diminished in communities with OD. For plotting these complex relationships, we used the Stata "margins" package. Margins calculates the predicted level of the outcome for specific points through the exposure distribution (OD) separately for dewormed and non-dewormed children, holding all other variables in the model constant. We predicted marginal effects at 10% increments from 0 to 100% OD.

We performed 2 sets of sensitivity analyses. Firstly, to fit a "within sanitation context" model, separate regressions were run for each outcome for the 3 OD subsamples (OD free, moderate OD, and high OD) to gauge changes in regression coefficients by varying OD context. Secondly, we ran models using continuous outcomes—Hb and serum levels of micronutrient status—using ordinary least squares to observe changes over the entire range of the outcomes, rather than at deficiency thresholds. Micronutrient status outcomes were log-transformed in these regressions to correct for long tails in their distributions at normal scales. In all regressions, for all outcomes, only complete cases were analyzed.

This study is reported as per the Strengthening the Reporting of Observational Studies in Epidemiology (STROBE) guideline (S1 Checklist).

## Results

### Sample summary

Of the 105,060 children in the sample, 48.0% were female with a mean age of 8.6 years (Table 1). Educational attainment was low among fathers (7.7 years) and mothers (5.3 years). On average, children consumed 4.7 out of 5 plant-sourced foods, 3.2 out of 4 animal-sourced foods, and 3.4 out of 4 unhealthy food groups per week. Household access to piped water, toilets, and treated water was 42.3%, 80.9%, and 26.9%, respectively. About 21.7% of households were surveyed in summer, 42.4% during the monsoon season, and 12.1% post monsoon, and the remaining during winter. Half the sample lived in OD free communities, a quarter in moderate OD, and the remaining quarter in communities with high OD (Table 1). Details of number of participants with valid (non-missing and within appropriate range) data for outcomes and covariates are shown in S1 Fig.

**Table 1. Summary statistics of sociodemographic and dietary characteristics of Indian children aged 1–19 years.**

| | Mean/Percent | 95% CI |
|---|---|---|
| **Exposures** | | |
| Consumed deworming medication, % | 43.4 | [43.1, 43.7] |
| No open defecation in community[1], % | 50.9 | [50.6, 51.2] |
| Moderate open defecation in community (>0< = 30%), % | 25.5 | [24.3, 24.8] |
| High open defecation in community (>30%), % | 25.6 | [24.3, 24.8] |
| **Covariates** | | |
| Child age, months | 103.4 | [103.0, 103.8] |
| Female, % | 48.0 | [47.7, 48.3] |
| Father's education, years | 7.7 | [7.7, 7.7] |
| Mother's education, years | 5.3 | [5.3, 5.4] |
| Count of plant sourced foods consumed per week out of 5, mean[2] | 4.7 | [4.7, 4.7] |
| Count of animal sourced foods consumed per week out of 4, mean[3] | 3.2 | [3.2, 3.2] |
| Count of unhealthy foods consumed per week out of 4, mean[4] | 3.4 | [3.4, 3.4] |
| Wealth quintile 1, poorest[5] | 20.0 | [19.8, 20.3] |
| Wealth quintile 2 | 20.0 | [19.8, 20.3] |
| Wealth quintile 3 | 20.1 | [19.8, 20.3] |
| Wealth quintile 4 | 20.0 | [19.7, 20.2] |
| Wealth quintile 5, nonpoor | 19.9 | [19.7, 20.2] |
| Household has piped water, % | 42.3 | [42.0, 42.6] |
| Household has toilet, % | 80.9 | [80.7, 81.2] |
| Household treats water, % | 26.9 | [26.7, 27.2] |
| Survey season | | |
| Summer (March, April, May), % | 21.7 | [21.5, 22.0] |
| Monsoon (June, July, August, September), % | 42.4 | [42.1, 42.7] |
| Post monsoon (October, November), % | 12.1 | [11.9, 12.3] |
| Observations | 105,060 | |

[1]Open defecation was measured as the average proportion of sampled households practicing open defecation in a community.

[2]Plant foods include legumes, leafy greens, vegetables, fruits, and nuts/seeds.

[3]Animal foods include eggs, fish, dairy, and poultry.

[4]Unhealthy foods include fried foods, junk food, sweets, and sugar-sweetened beverages.

[5]Wealth quintiles were constructed from a principal component analysis of 26 household assets, cooking fuel, house and land ownership, and house floor, roof, and wall materials. Water treatment includes boiling or filtering water with an electric filter.

## Deworming coverage and association with outcomes

Average consumption of deworming was 43.4% [IQR 26.0; 59.0] and among all age groups (**Table 1**). There was an inverse U-shaped relationship between age and the receipt of deworming medication (**Fig 1**). For children aged 1 to 4 years, the primary source of deworming medication was the Anganwadi Center or the frontline health worker, whereas older children mostly received deworming at school. Communities with higher OD had lower consumption of deworming (**S2 Fig**).

Compared to non-dewormed children, dewormed children had lower prevalence of anemia (22.4% versus 27.7%) (**Table 2**), iron deficiency (24.1% versus 26.0%), folate deficiency (27.4% versus 34.8%), and vitamin B12 deficiency (13.7% versus 17.9%). No significant differences were observed between dewormed and non-dewormed children for vitamin A deficiency (16.4%) and zinc deficiency (31.4%).

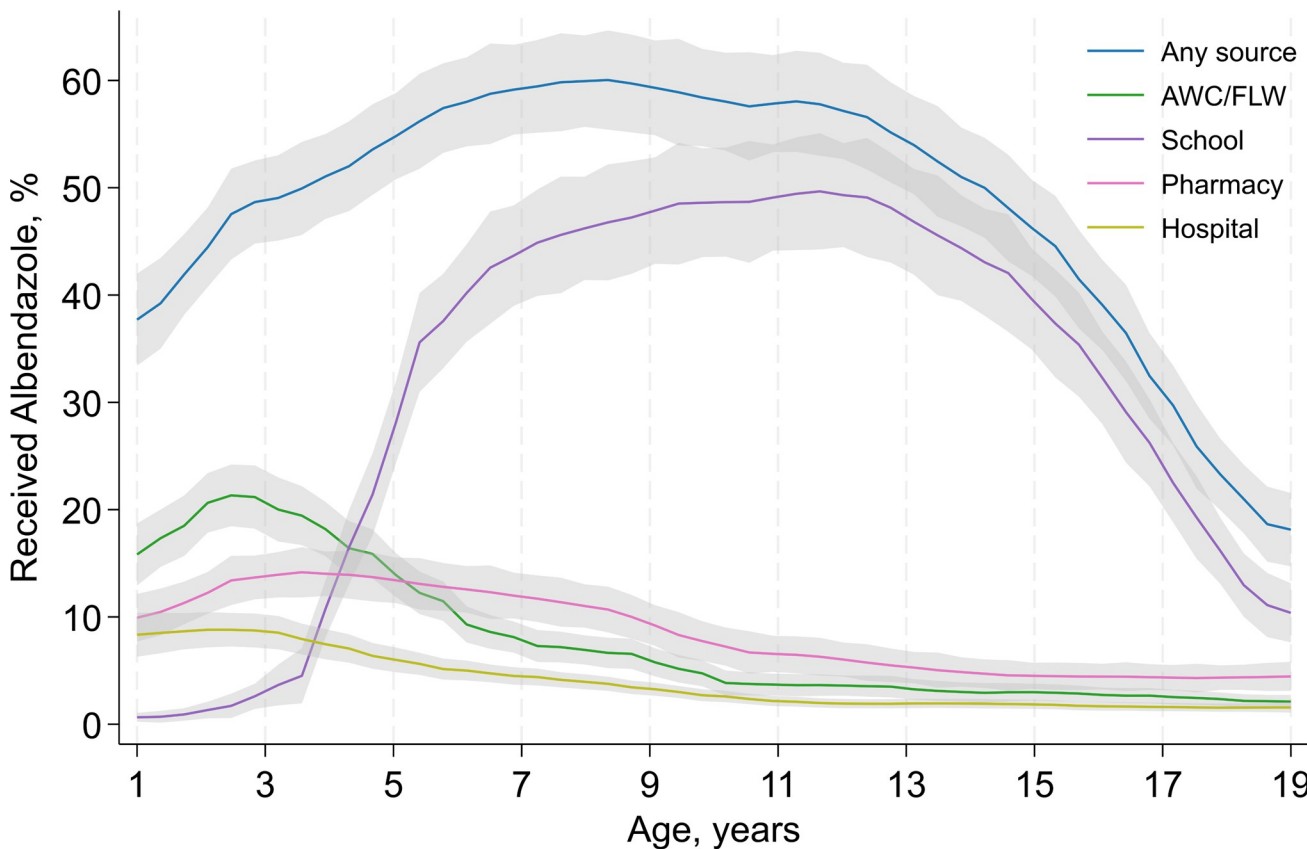

**Fig 1. Source of deworming medication among Indians aged 1–19 years by age in the 6 months preceding the survey.** AWC, Anganwadi center; FLW, frontline health worker.

## Association between deworming and nutritional status is modified by open defecation

Anemia, vitamin A, folate, and vitamin B12 deficiencies were greatest in communities with high OD and higher in communities with moderate OD compared to OD-free communities (unadjusted comparisons) (S2 Table). S3 Table shows results from the fully adjusted logit regression model examining interactions between deworming and cluster OD. In communities with no OD (cluster OD = 0), dewormed children had significantly lower odds of being

**Table 2. Prevalence of anemia and micronutrient deficiencies among Indians aged 1–19 years by deworming status.**

|  | Dewormed | | Not dewormed | | All | |
|---|---|---|---|---|---|---|
|  | % | 95% CI | % | 95% CI | % | 95% CI |
| Anemia (N = 41,431) | 22.4 | [21.8, 23.0] | 27.7 | [27.1, 28.3] | 25.3 | [24.9, 25.8] |
| Iron deficiency (N = 37,094) | 24.1 | [23.4, 24.7] | 26.0 | [25.4, 26.6] | 25.1 | [24.7, 25.6] |
| Zinc deficiency (N = 33,962) | 33.6 | [32.9, 34.3] | 33.6 | [32.9, 34.3] | 33.6 | [33.1, 34.1] |
| Vitamin A deficiency (N = 32,272) | 16.5 | [15.9, 17.1] | 16.3 | [15.8, 16.9] | 16.4 | [16.0, 16.8] |
| Folate deficiency (N = 40,661) | 27.4 | [26.7, 28.0] | 34.8 | [34.2, 35.4] | 31.4 | [31.0, 31.9] |
| Vitamin B12 deficiency (N = 34,070) | 13.7 | [13.1, 14.2] | 17.9 | [17.3, 18.4] | 16.0 | [15.6, 16.4] |

Note: See S1 Table for definitions of outcome variables.

anemic odds ratio (OR) = 0.71; (95% CI [0.66, 0.76], $p < 0.001$), iron deficient OR = 0.78; (95% CI [0.74, 0.82], $p < 0.001$), folate deficient OR = 0.69; (95% CI [0.64, 0.74], $p < 0.001$), and vitamin B12 deficient OR = 0.91; (95% CI [0.83, 0.99], $p < 0.001$), relative to non-dewormed children. Among non-dewormed children (dewormed = 0), maximum cluster OD (cluster OD = 1) was associated with significantly higher odds of anemia OR = 1.24; (95% CI [1.06, 1.46], $p = 0.003$), vitamin A deficiency OR = 1.55; (95% CI [1.20, 2.00], $p < 0.001$), and Vitamin B12 deficiency OR = 1.89; (95% CI [1.54, 2.34], $p < 0.001$). Cluster OD significantly modified the association between deworming and micronutrient deficiencies, revealed by significant coefficients on interaction terms for anemia, iron, vitamin A, and folate deficiencies. The predicted marginal effects from these interactions are shown in **Fig 2**, revealing that, across outcomes, deworming begins to become ineffective when approximately 30% to 60% of households in a community practice OD. Beyond this level, the confidence intervals of the predicted margins among dewormed individuals overlap with non-dewormed individuals for anemia, iron, and folate deficiency. Moreover, folate deficiencies were higher among dewormed children, but the predicted margins had very high uncertainty at high OD levels.

## Sensitivity analyses

**Fig 3** shows results from the fully adjusted logit regression model examining associations between deworming and micronutrient deficiencies in varying OD contexts (subsamples of OD intervals). In communities with no OD, dewormed children had significantly lower odds of being anemic OR = 0.72; (95% CI [0.66, 0.78], $p < 0.001$), iron deficient OR = 0.73; (95% CI [0.68, 0.78], $p < 0.001$), and folate deficient OR = 0.65; (95% CI [0.59, 0.72], $p < 0.001$) relative to non-dewormed children. However, OD context modified the association between deworming and micronutrient deficiencies. Deworming was associated with the greatest reduction in odds of anemia in no OD communities, followed by moderate OD communities OR = 0.78; (95% CI [0.71, 0.87], $p < 0.001$) while the smallest reduction was in high OD communities OR = 0.84; (95% CI [0.76, 0.92], $p < 0.001$) although the differences were not statistically significant. An attenuation of dose response was also observed for iron, folate deficiency, and Vitamin A, moving the odds ratio on deworming closer to the null with increasing OD, where the difference in deficiency following deworming in no OD communities versus high OD communities was statistically significant for iron and folate. In no OD communities, vitamin A deficiencies were lower among dewormed children OR = 0.92; (95% CI [0.83, 1.01], $p = 0.100$). Deworming was not significantly associated with zinc and vitamin B12 deficiency in any OD context.

In our second sensitivity analysis using micronutrient outcomes as continuous variables, in communities with no OD, dewormed children had 0.16 points higher log Hb g/dl, 0.08 points higher log ferritin µg/l, 0.09 points higher log folate ng/ml, and 0.04 points higher log vitamin B12 pg/ml (all $p < 0.05$), than non-dewormed children (**S3 Fig**). These associations were attenuated for anemia and were no longer statistically significant for log ferritin, log folate, and log vitamin B12 for dewormed versus non-dewormed children in moderate and high OD contexts.

## Discussion

We used nationally representative data from India to investigate the potential influence of community sanitation levels, measured by the percentage of households in a community practicing open defecation, on the efficacy of deworming interventions in diminishing the prevalence of anemia and micronutrient deficiencies among children and adolescents in India. Our analyses reveal 3 salient findings. Firstly, deworming coverage in India is low, especially

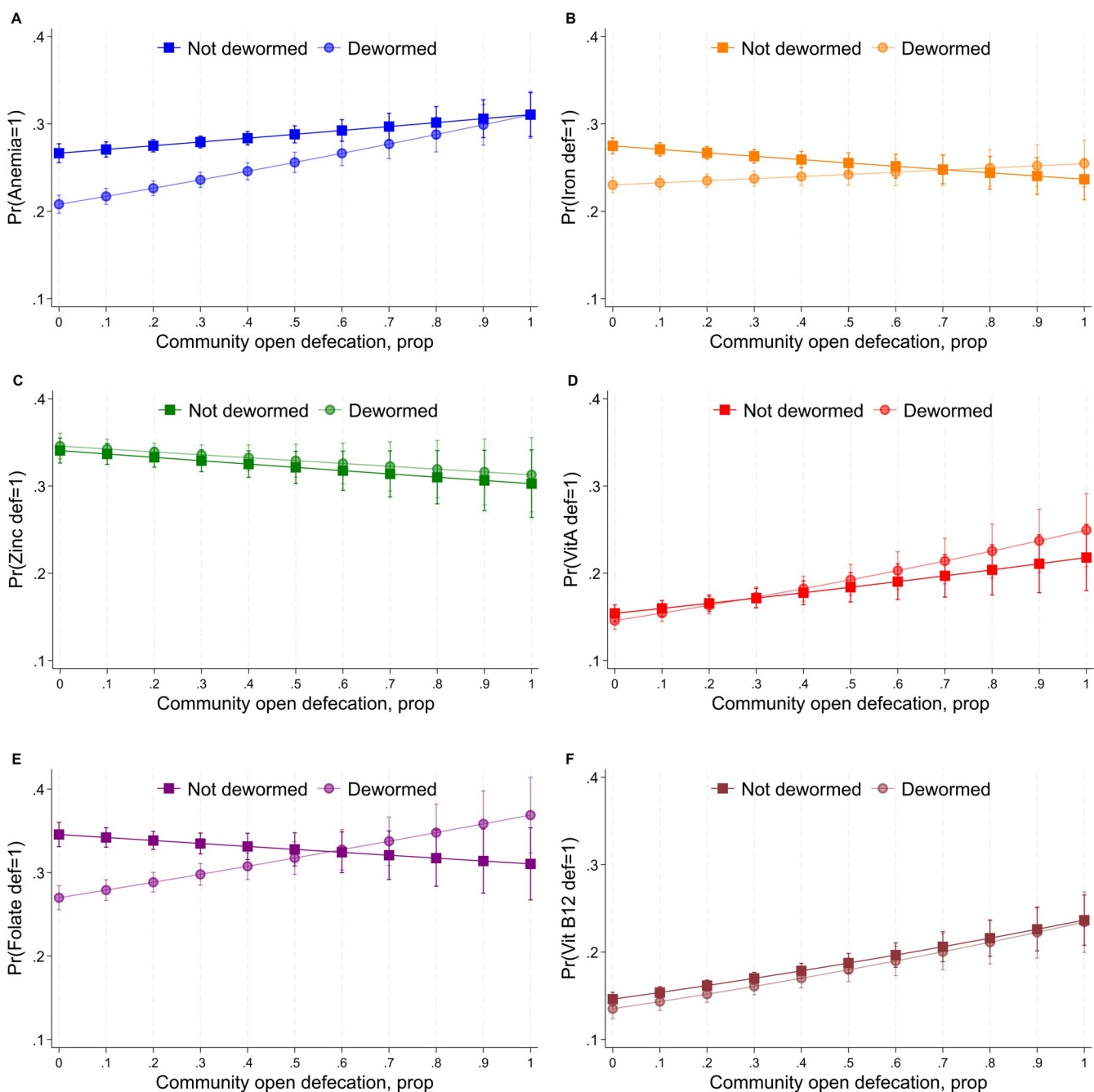

**Fig 2. Probability of anemia or micronutrient deficiencies for dewormed vs. non-dewormed individuals by continuous community open defecation level among Indians aged 1–19 years.** Squares and dots are means and bands are confidence intervals of predicted probabilities from logistic regression models using the "Margins" command in STATA 17. Regression models are adjusted for sex, age, parental education, wealth, dietary patterns, and seasonality. Standard error estimates are clustered at the PSU level. OD was measured as the average proportion of sampled households practicing open defecation in a community. OD, open defecation; PSU, primary sampling unit.

among young children and adolescents out of school. Secondly, however, OD prevalence modifies the association between deworming and micronutrient deficiencies in India. Thirdly, deworming becomes ineffective at addressing micronutrient deficiency and anemia at a threshold range of 30% to 60% OD.

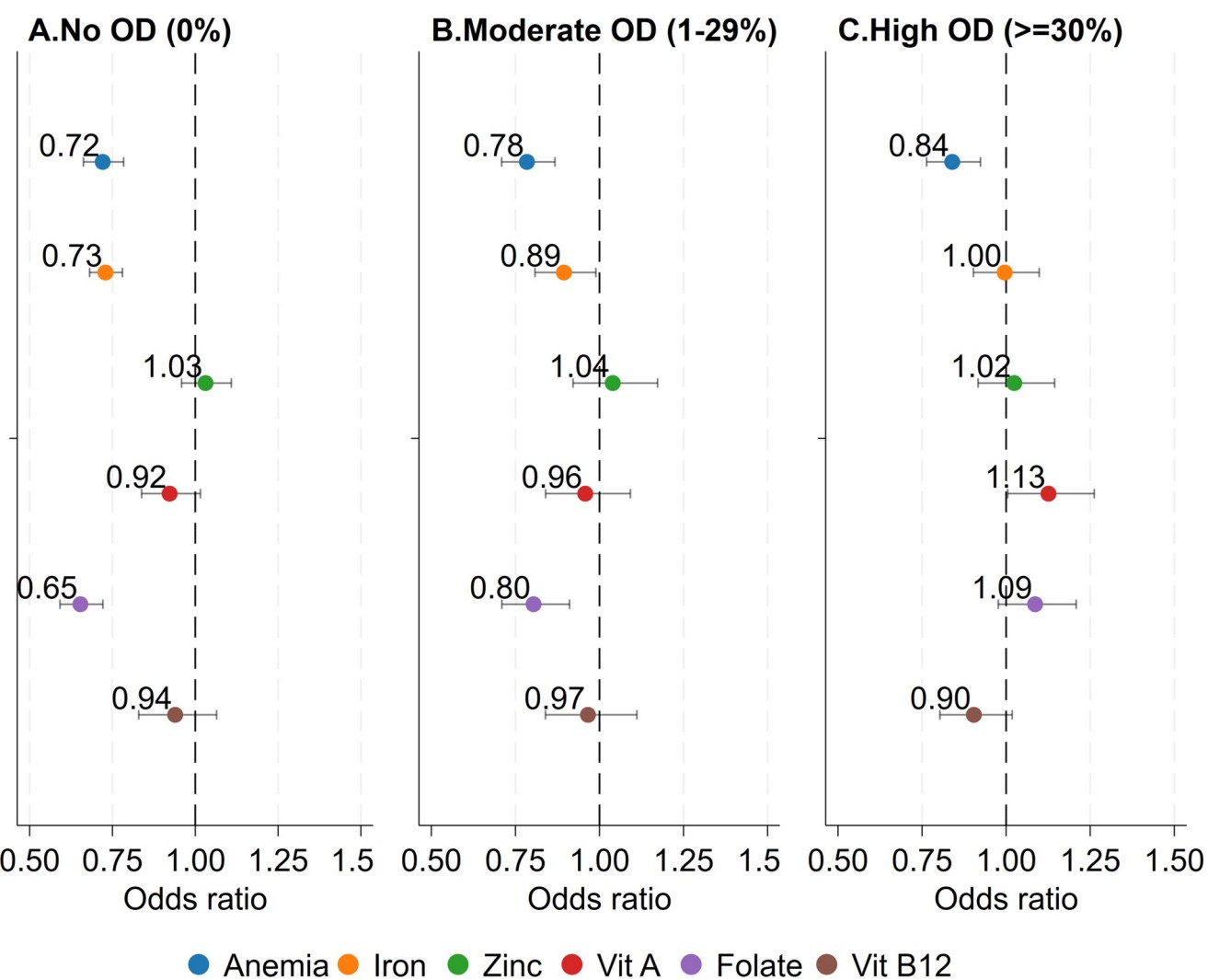

**Fig 3. Odds of anemia or micronutrient deficiencies for dewormed vs. non-dewormed individuals by low, moderate, and high open defecation levels among Indians aged 1–19 years.** ORs are interpreted as the odds of being anemic or micronutrient deficient among children who were dewormed relative to those who were not. Logistic regression models are adjusted for sex, age, parental education, wealth, dietary patterns, and seasonality. Standard error estimates are clustered at the PSU level. OD was measured as the average proportion of sampled households practicing open defecation in a community, then was divided in 3 categories: OD free, moderate open defecation (1%–29% of households in a community practicing OD), high open defecation (30% to 100% of households in a community practicing OD). OD, open defecation; OR, odds ratio; PSU, primary sampling unit.

Anemia and deficiencies in essential micronutrients hinder human capital development in LMICs [46]. While STH infections, anemia, and impaired micronutrient absorption are inter-related, deworming interventions have shown limited effectiveness in mitigating these adverse outcomes in generalizable findings across multiple contexts [6–8]. This raises the possibility that sanitation is a necessary precursor for deworming interventions in addressing the burdens associated with STH infections [9]. To the best of our knowledge, our study is the first to empirically demonstrate that targeted deworming interventions may be less successful in OD prevalent areas. The regression analyses indicate that deworming interventions effectively mitigate anemia prevalence, as well as deficiencies in iron and folate, within Indian communities that are free from OD. However, the effectiveness of deworming considerably diminishes in communities characterized by moderate to high levels of OD (>30%). Prior studies have also

suggested critical thresholds at which community-wide sanitation significantly affects nutrition outcomes [47,48]. For example, a recent investigation identified a range of 25% to 50% OD prevalence as being pivotal for child height across 4 countries [49]. In this study, height gains were only realized once communities reached OD levels below the critical threshold and continued until communities were OD free. Another study reported an inflection in the OD dose-response function for anemia prevalence in India, suggesting anemia reduction benefits are only realized once communities reach the critical OD threshold of 40% [50]. This underlines the importance of explicitly modeling community-wide environmental externalities of OD in the Indian and global contexts.

Our study's strengths include the utilization of nationally representative data, the objective measurement of nutritional and health outcomes, and adjustment for multiple covariates to mitigate the influence of confounding factors. Unlike all Demographic Health Surveys, which have limited data on only anemia among children aged under 5, the CNNS provides a wide range of micronutrient status data for children aged 1 to 19 years. Our findings are policy relevant, particularly for India and other LMICs that currently confront the challenges posed by STH burdens.

Our study bears certain limitations. Firstly, owing to the cross-sectional nature of our data, our analysis primarily establishes associations rather than causation, as it could not track changes in outcomes in response to variations in exposures over time. Biases in observational cross-sectional studies can arise due to unmeasured omitted confounders including variables that exhibit correlation with deworming, micronutrient status, and OD, that were not included in our study. For example, some communities may have poor governance leading to poor health and sanitation services. Secondly, deworming status was self-reported by respondents and OD was estimated from observations on a sample of households from the community thereby creating the potential for measurement error which may bias our estimates towards the null. Thirdly, due to the absence of data regarding the timing of deworming medication consumption and the measurement of STH infections, we were unable to explore the relationship between the frequency of deworming interventions and STH infections in the context of varying OD levels. This presents the possibility that the administration of 2 doses of deworming per year in OD-affected communities may be insufficient to effectively counteract STH exposure. Therefore, the consideration of more frequent dosing regimens or extending deworming to all age groups merits investigation [51]. We encourage future research endeavors to adopt a longitudinal approach, allowing for the examination of changes in micronutrient status and STH infections in response to alterations in deworming frequency and dosage within evolving OD contexts. Finally, the dataset did not provide indicators for access to micronutrient supplementation among children and adolescents, thus precluding an investigation of interactions of such interventions with deworming and sanitation.

Cross-sectional studies are subject to systematic errors or biases, and the association between WASH and STH is not consistently supported by the available evidence from randomized controlled trials [52]. Conventional approaches often focus on low-cost household interventions, neglecting broader fecal exposure risks linked to undernutrition [47]. Moreover, to estimate the causal interaction effects of deworming on micronutrient status in different community sanitation contexts would require researchers to randomize deworming treatments and sanitation levels across communities [53]. In practical terms, such a study would be very difficult to implement. Thus, it is prudent to rely on large-scale observational datasets to inform programs and policies until such trials are conducted.

Our findings have implications for the implementation of deworming and WASH programs and policies in LMICs, considering the WHO roadmap for neglected tropical diseases from 2021 to 2030, as we discuss in the following section. The pillars include (1) accelerating

programmatic action; (2) intensifying cross-cutting approaches; and (3) changing operating models and culture to facilitate country ownership [54]. Concerning the first pillar of the WHO's roadmap, which focuses on expediting programmatic action, the data suggest that deworming coverage remains notably inadequate in India, leaving considerable scope for enhancement and expansion of deworming interventions. Despite national initiatives to scale up deworming, only 41% of children aged 12 to 36 months were given deworming as per the fifth round (2019 to 2021) of India's National Family Health Survey [55]. To achieve higher coverage, it is imperative to place a stronger emphasis on the second pillar of the strategy, which involves the intensification of cross-cutting approaches. For example, evidence from LMICs suggests that where strong health-care services exist, such as routine health care and public health programs, deworming coverage is likely to be higher because deworming can be readily delivered using existing services [56]. Further, the strategic utilization of village health centers and educational institutions as effective delivery platforms for deworming interventions is needed. We also found low coverage, especially in children under the age of 5 who are targeted through community-level Anganwadi centers. Deworming coverage significantly improves as children progress into primary and secondary school. Nevertheless, even among primary and secondary school attendees, deworming coverage hovers around 60% and declines as children reach the age of 14 years. This observation suggests that a significant proportion of higher secondary school students may not be receiving deworming medication or adolescents who are not enrolled in educational institutions lack alternative avenues for accessing deworming interventions. Moreover, consumption of deworming medication is lower, on average, in communities where the prevalence of OD is high. This inverse relationship between OD and deworming levels is likely attributable to weaker public health services in such communities, suggesting a need to identify such hotspots and prioritize improvements in sanitation services, messaging to reduce OD, and the distribution of deworming pills at schools, Anganwadi centers, and by frontline health workers in such areas.

The results align with the findings of cluster-randomized trials conducted in Uttar Pradesh, a state in northern India where OD is common, which found insignificant reductions in child hemoglobin by deworming [57]. Our findings suggest that null impacts are likely attributable to recurrent exposure of individuals to STH infections in densely populated communities where OD is practiced and also depends on worm species involved [58]. Considering the Indian government's multifaceted approach, encompassing numerous programs across various ministries with substantial investments directed toward deworming and WASH initiatives, our findings underscore the potential necessity to monitor coverage of deworming programs alongside WASH indicators though regular surveys in the same communities [59–61]. This will enable appropriate sequencing of deworming events (e.g., National Deworming Day) with BCC campaigns for improved sanitary practices. Furthermore, such convergent actions should be targeted at high STH and OD burden communities to avert reinfections and prevent a resurgence of STH-related morbidity in high-risk areas following deworming interventions [22,62,63].

STH prevalence is influenced by climatic conditions (especially land surface temperature, humidity) and soil structure that favor embryonation of ova (Ascaris and Trichuris) and larval development and survival (hookworms) [64]. STH exhibit considerable geographic heterogeneity in distribution in India, both due to these climatic and soil differences and due to socio-demographic, migration, and hygiene behavioral patterns as well as previous rounds of deworming [58]. STH are also known to vary at finer spatial scales of village or community within districts and have been linked to poor household WASH indicators in nearly all surveys [65]. Surveys from India have indicated higher levels of Ascaris in north and northeastern India and in urban populations and higher levels of hookworm infections in southern India

and in rural populations in some studies, although the reliability of these wide ranging estimates are affected by the populations surveyed (age groups included, sampling at health centers or schools) and quality of stool screening carried out (methodology used and whether screened rapidly enough to ensure detection of hookworm ova) [58,66,67]. Our study underscores the pressing requirement for more detailed assessments of the burdens associated with distinct worm species at finer geographic scales within India. This need further arises from existing evidence suggesting that albendazole, the most employed anthelmintic medication, exhibits inefficacy in combatting the burden posed by whipworms [68]. Additionally, it is imperative to estimate the national and subnational burdens of roundworm, whipworm, and hookworm infections and their associations with water sources, fecal contamination, and OD practices in communities across India.

The extent that improved sanitation is practiced in a community, and therefore, the degree of OD, exerts a significant influence on the relationship between deworming interventions and the anemia and micronutrient status of children and adolescents aged 1 to 19 years in India. Expanding the coverage of deworming programs, alongside a concurrent reduction in OD rates, may hold the potential to significantly alleviate the burden of STH infections in India. However, in circumstances of high OD (more than 60%), deworming appears to become ineffective in addressing anemia and micronutrient deficiencies in Indian children. This suggests that the public health system should consider sequencing deworming interventions, with focused targeting in communities that have already achieved minimum thresholds of OD of at least 30% and sequencing sanitation interventions in high OD communities before deworming days, ensuring high coverage for both. Interventions to reduce OD, and keep it low, might include community-led triggering and retriggering events and social marketing to support local latrine manufacture and construction [69], and maintenance efforts to extend the lives of the pits—and therefore latrine use—such as the construction of double-pits where space permits, subsidies for pit emptying and vermifiltration [70]. Participation in deworming and therefore coverage of targeted populations is influenced by several factors such as gender, socioeconomic status, population density, school attendance, local religious and cultural beliefs, awareness of helminth infections, and trust in public health campaigns [71,72]. Community engagement and sensitization activities need to be tailored to address these factors in the form of media announcements, social media, or local dissemination by trained and motivated community drug distributors (including schoolteachers) prior to national deworming campaigns [73,74]. Furthermore, sanitation and deworming alone are insufficient to address micronutrient deficiencies among children in India. More research is needed on the efficacy of micronutrient supplementation for children and adolescents as a complementary strategy for improving nutritional outcomes alongside deworming and OD elimination.

## Supporting information

**S1 Checklist. Strengthening the Reporting of Observational Studies in Epidemiology (STROBE).**
(DOCX)

**S1 Fig. Strengthening the Reporting of Observational Studies in Epidemiology (STROBE) sample sizes flow diagram.**
(DOCX)

**S2 Fig. Consumption of deworming medication in the previous 6 months among Indians aged 1–19 years by community open defecation level.**
(DOCX)

**S3 Fig. Coefficients from regression of biomarkers on deworming status by community open defecation level among Indians aged 1–19 years.**
(DOCX)

**S1 Table. Biomarker measurement and deficiency thresholds for outcomes.**
(DOCX)

**S2 Table. Prevalence of anemia and micronutrient deficiencies among Indians aged 1–19 years by open defecation level.**
(DOCX)

**S3 Table. Association between deworming, community OD, and their interaction with anemia and micronutrient deficiencies in India children.**
(DOCX)

**S1 Dataset. Supporting dataset.**
(ZIP)

## Author Contributions

**Conceptualization:** Suman Chakrabarti, Avinash Kishore, Samuel Scott.

**Data curation:** Suman Chakrabarti.

**Formal analysis:** Suman Chakrabarti, Phuong H. Nguyen.

**Funding acquisition:** Sitara S. R. Ajjampur.

**Investigation:** Suman Chakrabarti, Sitara S. R. Ajjampur, Hugh Sharma Waddington, Avinash Kishore, Samuel Scott.

**Methodology:** Suman Chakrabarti, Sitara S. R. Ajjampur, Hugh Sharma Waddington, Avinash Kishore, Phuong H. Nguyen, Samuel Scott.

**Software:** Suman Chakrabarti.

**Supervision:** Avinash Kishore, Samuel Scott.

**Visualization:** Suman Chakrabarti.

**Writing – original draft:** Suman Chakrabarti, Sitara S. R. Ajjampur, Hugh Sharma Waddington, Avinash Kishore, Phuong H. Nguyen, Samuel Scott.

**Writing – review & editing:** Suman Chakrabarti, Sitara S. R. Ajjampur, Hugh Sharma Waddington, Avinash Kishore, Phuong H. Nguyen, Samuel Scott.

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
