## [Editor Report · Decision Letter 0]

8 Feb 2024

Dear Dr Scott, 

Thank you for submitting your manuscript entitled "THE UTILITY OF DEWORMING TO ADDRESS ANEMIA AND MICRONUTRIENT DEFICIENCIES IN COMMUNITIES THAT PRACTICE OPEN DEFECATION: NATIONALLY REPRESENTATIVE EVIDENCE FROM INDIA" for consideration by PLOS Medicine.

Your manuscript has now been evaluated by the PLOS Medicine editorial staff and I am pleased to let you know that we would like to send your submission out for external peer review.

Please re-submit your manuscript within two working days, i.e. by Feb 12 2024 11:59PM.

Kind regards,

Katrien G. Janin, PhD

Senior Editor

PLOS Medicine

---

## [Decision Letter · Decision Letter 1]

28 Feb 2024

Dear Dr. Scott,

Thank you very much for submitting your manuscript "THE UTILITY OF DEWORMING TO ADDRESS ANEMIA AND MICRONUTRIENT DEFICIENCIES IN COMMUNITIES THAT PRACTICE OPEN DEFECATION: NATIONALLY REPRESENTATIVE EVIDENCE FROM INDIA" (PMEDICINE-D-24-00428R1) for consideration at PLOS Medicine. 

[LINK]

After discussing the paper with the editorial team and an academic editor with relevant expertise, I’m pleased to invite you to revise the paper in response to the reviewers’ comments. We plan to send the revised paper to the original reviewers, and we cannot provide any guarantees at this stage regarding publication. 

When you upload your revision, please include a point-by-point response that addresses all of the reviewer and editorial points, indicating the changes made in the manuscript and either an excerpt of the revised text or the location (eg: page and line number) where each change can be found. Please submit a clean version of the paper as the main article file and a version with changes marked should as a marked-up manuscript. Please also check the guidelines for revised papers at http://journals.plos.org/plosmedicine/s/revising-your-manuscript for any that apply to your paper. 

We look forward to receiving your revised manuscript. Please do not hesitate to contact me (kjanin@plos.org) with any questions you may have 

Sincerely,

Katrien Janin, PhD

PLOS Medicine

plosmedicine.org

Comments from the reviewers:

Reviewer #1: I am grateful to have been afforded the opportunity to review this manuscript. Despite its potential importance to the readers, this manuscript requires revisions. Below are a few considerations the authors could take into account in order to fortify the manuscript.

Title: Enhancing the clarity and relevance of the title: The title should be direct, accurate, interesting, concise, precise and unique. 

Abstract:

1. The abstract could be improved by condensing highlighting key content areas and eliminating irrelevant details.

2. Authors should consider incorporating a discussion to include implications for policy and public health interventions. 

Introduction

1. Line 57: The citation is missing, please cite 

2. It may be beneficial to provide more background information on the relation between STH exposure and anemia or micronutrient deficiencies to better understand the hypothesis being tested.

Methods

1. Line 117: The authors should clarify whether this study employed secondary data analysis

2. The author should include more information on how the samples were handled and processed. 

Results

1. Table 1: Superscript 5 is missing in the note

2. Authors should provide an overview of the potential confounding variables that may have affected the outcome of their study

Discussion

1. The authors should consider expanding on the implications of their findings and discussing potential strategies to address deworming coverage and high OD prevalence

General comment

The manuscript exhibits promising potential to captivate its readers. Nevertheless, it requires minor revisions for optimal refinement

Reviewer #2: See attachment

Michael Dewey

Reviewer #3: I would like to congratulate the authors on this valuable paper. This provides comprehensive information from a very big national survey on the range of nutrition interactions between open defecation, deworming and infections. This is crucial for policy making around WASH interventions. I have a few very minor comments here. But, overall the paper is very good. 

Line 57 - There is a marker for a reference that you have forgotten to put in. 

Line 64 - I don't understand the logic of this last sentence. I think there is the need to consider sanitation interventions ,as well as behaviour change communication as your results show.

Line 78-80 - Please include a reference for the first two sentences here regarding the numerous factors contributing to STH in INdia. 

Line 108 - I think this sentence would be more effective if you included the direction of the association between deworming and outcomes. For example: we hypothesise the association between deworming and reductions in anemia or micronutrient deficiencies....

Line 125 - survey conducted in 30 out of how many states in India? will give us a better picture of the scale of the survey.

Line 157 - You could describe in a bit more detail how the nutrition information was collected. ie food frequency questionnaire? using a validated instrument?

Line 157 - I don't understand the dietary diversity variable. You have not used a standard procedure for assessing dietary diversity, and it seems you include 'unhealthy' foods as one of the components of dietary diversity. Could you explain a bit more why you use this approach? And in the Results you don't actually seem to use this dieatry diversity measure, you just report separately for each food type. 

Finally Table 2 is missing from the file i received. I could basically understand from the Results text what was there but i need to see the Table as well. 

Reviewer #4: Chakrabarti and colleagues present an interesting secondary analysis of India's national nutrition survey to investigate the relationships between open defecation (OD), deworming, and nutritional status. The analyses show that the impact of deworming on micronutrient status is modified by the level of OD in the community. The analysis is appropriate for the data and the interpretation is sound. However, there are several issues that would benefit from careful consideration, including the following issues.

Numerous studies have reported associations between the risk of STH and access to WASH. However, cross-sectional studies are subject to systematic errors or biases, and the association between WASH and STH is not consistently supported by the available evidence from randomized controlled trials. These trials report varied effects of WASH interventions on infection risk. The potential reasons for the incongruence between cross-sectional studies and trials are discussed in more detail in the following viewpoint: https://parasitesandvectors.biomedcentral.com/articles/10.1186/s13071-019-3532-6

The relationship between STH, micronutrient status and OD is likely to differ depending on the specific STH species, owing to differences in life cycle and morbidity. While the available data do not permit a species-level analysis, it is essential for the discussion to address this aspect. This should include a discussion of the geographical distribution of each species in India, as the results might vary by geography. 

Although the results are intuitive, their accuracy relies on the potential for reporting bias in deworming and OD. This aspect requires thorough discussion.

The cross-sectional design of the analysis introduces the possibility of unobserved confounding, and it would be important to consider potential confounders in the discussion.

This pertains to the style of the journal, but many results are presented both within the text and in tables, and there is potential to minimize redundancies in reporting.

It would be useful to specify the coefficients of the polynomial smoothing used to develop age profiles of deworming and OD.

[LINK]

Editorial comments and general Journal request:

1. Please revise your title according to PLOS Medicine's style. Your title must be nondeclarative and not a question. It should begin with main concept if possible. "Effect of" should be used only if causality can be inferred, i.e., for an RCT. Please place the study design ("A randomized controlled trial," "A retrospective study," "A modelling study," etc.) in the subtitle (ie, after a colon).

2. Please structure your abstract using the PLOS Medicine headings (Background, Methods and Findings, Conclusions). In the last sentence of the Abstract Methods and Findings section, please describe the main limitation(s) of the study's methodology.

3. Please ensure that the study is reported according to the STROBE guideline, and include the completed STROBE checklist as Supporting Information. Please add the following statement, or similar, to the Methods: ""This study is reported as per the Strengthening the Reporting of Observational Studies in Epidemiology (STROBE) guideline (S1 Checklist).The STROBE guideline can be found here: http://www.equator-network.org/reporting-guidelines/strobe/

When completing the checklist, please use section and paragraph numbers, rather than page numbers."

4. Discussion: please present and organize the Discussion as follows: a short, clear summary of the article's findings; what the study adds to existing research and where and why the results may differ from previous research; strengths and limitations of the study; implications and next steps for research, clinical practice and/or public policy implications; followed by a one-paragraph conclusion. Please remove all subheadings within your Discussion.

5. References: Please use the "Vancouver" style for reference formatting, and see our website for other reference guidelines https://journals.plos.org/plosmedicine/s/submission-guidelines#loc-references. 

6. For in-text reference, citations are placed within square parentheses and should precede punctuation. Please amend throughout.

7. Where you provide CI values, please also provide p values for all results where appropriate (including the abstract), check and amend throughout. We suggest reporting statistical information in the following format: ‘x’; (95% CI [‘y’,’ z’] p value). We suggest use of commas as opposed to hyphens (as these can be confused with negative values) to separate upper and lower bounds. For p values, please report as p<0.001 and where higher as 'p=0.002'. Please add the statistical method used to your method section.

We expect to receive your revised manuscript by Mar 20 2024 11:59PM. Please email us (plosmedicine@plos.org) if you have any questions or concerns.

---

## [Decision Letter · Decision Letter 2]

5 Apr 2024

Dear Dr. Scott,

Thank you very much for re-submitting your manuscript "DEWORMING AND MICRONUTRIENT STATUS BY COMMUNITY OPEN DEFECATION PREVALENCE: A MODELLING STUDY FROM INDIA" (PMEDICINE-D-24-00428R2) for review by PLOS Medicine.

I have discussed the paper with my colleagues and the academic editor and it was also seen again by the reviewers. I am pleased to say that provided the remaining editorial and production issues are dealt with we are planning to accept the paper for publication in the journal.

[LINK]

We look forward to receiving the revised manuscript by Apr 12 2024 11:59PM.   

Sincerely,

Katrien Janin, PhD

Senior Editor 

PLOS Medicine

plosmedicine.org

Requests from Editors:

GENERAL: 

We suggest reporting statistical information in the following format: ‘x’; (95% CI [‘y’,’ z’] p value) and use commas as opposed to hyphens (as these can be confused with negative values) to separate upper and lower bounds. For p values, please report as p<0.001 and where higher as 'p=0.002'. Please add the statistical method used to your method section. We also invite you to report p values to consistently to the third decimal digit - thousandths.

Supplementary materials: Please note that supplementary materials are not checked and will be posted as supplied by the authors. Therefore, please double check. Please cite your Supporting Information as outlined here: https://journals.plos.org/plosmedicine/s/supporting-information - Please note you may use almost any description as the item name of your supporting information as long as it contains an "S" and number. For example, “S1 Appendix” and “S2 Appendix,” “S1 Table” and “S2 Table. Please ensure each supplementary material has a call out (link) from your main manuscript. 

To help us extend the reach of your research, please provide any Twitter handle(s) that would be appropriate to tag, including your own, your coauthors’, your institution, funder, or lab.

TITLE: 

We note that your use ‘modelling study’ as the study descriptor in your title, but your study is an observational study. We suggest you change your title to: “Deworming and micronutrient status by community open defecation: An observational study using nationally representative data from India, 2016-18.", or something similar.

DATA AVAILABILITY:

Thank you for providing all your data on acceptance. At that time, please ensure you update your data availability statement.

ABSTRACT:

Please structure your abstract using the PLOS Medicine headings: Background, Methods and Findings, Conclusions. Please note that Methods and Findings is one section. Please remove all other subheaders.

Abstract Methods and Findings:

In the opening line, please add that you mean “India” when your refer to National representation.

In the last sentence of the Abstract Methods and Findings section, please describe the main limitation(s) of the study's methodology.

AUTHORS SUMMARY:

Ideally each sub-heading should contain 2-3 single sentence, concise bullet points containing the most salient points from your study.

In the final bullet point of ‘What Do These Findings Mean?’ Please include the main limitations of the study in non-technical language.

Comments from Reviewers:

Reviewer #1: The authors have made extensive revisions and, upon thorough evaluation of the manuscript, it now satisfies the PLOS Medicine standards for publication.

Reviewer #2: The authors have addressed my points.

Michael Dewey

Reviewer #4: All my comments have been adequately addressed in the revised version. Thank you.

[LINK]

---

## [Editor Report · Decision Letter 3]

12 Apr 2024

Dear Dr Scott, 

On behalf of my colleagues and the Academic Editor, I am pleased to inform you that we have agreed to publish your manuscript "DEWORMING AND MICRONUTRIENT STATUS BY COMMUNITY OPEN DEFECATION PREVALENCE: AN OBSERVATIONAL STUDY USING NATIONALLY REPRESENTATIVE DATA FROM INDIA, 2016-18" (PMEDICINE-D-24-00428R3) in PLOS Medicine.

PRESS

Sincerely, 

Katrien G. Janin, PhD 

Senior Editor 

PLOS Medicine